# LoBE-GS: Load-Balanced and Efficient 3D Gaussian Splatting for Large-Scale Scene Reconstruction

## Abstract

3D Gaussian Splatting (3DGS) has established itself as an efficient representation for real-time, high-fidelity 3D scene reconstruction. However, scaling 3DGS to large and unbounded scenes such as city blocks remains difficult. Existing divide-and-conquer methods alleviate memory pressure by partitioning the scene into blocks, but introduce new bottlenecks: (i) partitions suffer from severe load imbalance since uniform or heuristic splits do not reflect actual computational demands, and (ii) coarse-to-fine pipelines fail to exploit the coarse stage efficiently, often reloading the entire model and incurring high overhead. In this work, we introduce LoBE-GS, a novel *Load-Balanced and Efficient* 3D Gaussian Splatting framework, that re-engineers the large-scale 3DGS pipeline. LoBE-GS introduces a depth-aware partitioning method that reduces preprocessing from hours to minutes, an optimization-based strategy that balances visible Gaussians—a strong proxy for computational load—across blocks, and two lightweight techniques, visibility cropping and selective densification, to further reduce training cost. Evaluations on large-scale urban and outdoor datasets show that LoBE-GS consistently achieves up to $2\times$ faster end-to-end training time than state-of-the-art baselines, while maintaining reconstruction quality and enabling scalability to scenes infeasible with vanilla 3DGS.

## 1 Introduction

Recent advances in 3D scene reconstruction and novel view synthesis have shifted from classical photogrammetry and Neural Radiance Fields (NeRFs) toward explicit, real-time representations. While photogrammetry offers geometric precision but poor rendering efficiency, NeRFs achieve photorealism but remain computationally expensive. 3D Gaussian Splatting (3DGS) addresses these limitations by representing scenes with millions of anisotropic Gaussian primitives optimized through a GPU-friendly rasterization pipeline, delivering both high fidelity and real-time performance. Its efficiency has quickly established 3DGS as a leading representation for scalable 3D content creation.

Despite its success in bounded scenes, scaling 3DGS to large and unbounded environments, such as city-scale reconstructions, remains an open challenge. The memory and computational costs scale with the number of Gaussian primitives, leading to optimization times and GPU usage that quickly become prohibitive. To mitigate this, recent works such as CityGaussian (CityGS) (Liu et al., 2025), VastGaussian (VastGS) (Lin et al., 2024), and DOGS (Chen & Lee, 2024) adopt a divide-and-conquer strategy, partitioning large scenes into spatial blocks that can be processed in parallel. While effective in reducing raw memory pressure, this paradigm introduces new bottlenecks as follows.

- **Lack of load balancing:** Current partitioning strategies do not explicitly account for computational load balance. Heuristics such as uniform grid splits or block size normalization often yield sub-regions with highly uneven optimization demands. As a result, the slowest block dominates the total training time, creating a long-tail bottleneck.

- **Inefficient coarse-to-fine pipelines:** Methods employing a coarse-to-fine pipeline, such as CityGS (Liu et al., 2025), fail to fully exploit the coarse stage for accelerating fine-level

optimization. The coarse model is typically reloaded in full, incurring heavy computational overhead.

To overcome these limitations, we introduce **LoBE-GS**, a novel framework that fundamentally re-engineers the large-scale 3DGS pipeline for load-balanced and efficient parallel training. LoBE-GS addresses the inefficiency of heuristic partitioning, improves the utilization of coarse models, and establishes a standardized evaluation protocol. We first introduce a novel partitioning approach that radically reduces the data partitioning time. Existing methods can result in a complex $O(M \times N)$ projection problem, where $M$ is the number of blocks and $N$ is the number of camera views, requiring up to several hours. Our method leverages depth information from a coarse model to assign each camera to its corresponding block with a single, highly efficient projection per camera. This reduces the projection complexity to a linear $O(N)$ time and shortens the preprocessing time from hours to minutes.

To avoid unbalanced loading in each block for the parallel training, we employee an optimization to scene partitioning that directly addresses the load-balancing problem. Our experiments revealed a strong correlation between the initial number of visible Gaussians in the blocks and the subsequent optimization time. We therefore adopt the number of visible Gaussians as a reliable *proxy for computational load*. By explicitly balancing this metric across blocks, our framework eliminates long-tailed training bottlenecks and ensures more uniform computational demands. Moreover, we propose two complementary techniques to reduce the computational load of each block. First, we introduce *visibility cropping*, a technique applied after scene partitioning to prune irrelevant Gaussians from each block, which reduces the training time without sacrificing the quality of the final reconstruction. Second, we propose *selective densification* to further reduce the computational load of each block by strategically adding or cloning Gaussians only when needed.

We evaluate LoBE-GS on diverse large-scale datasets, including urban and outdoor scenes spanning hundreds of meters. Experimental results show that our method consistently delivers faster training and more balanced computation than prior approaches, while maintaining or improving reconstruction quality. In particular, LoBE-GS reduces end-to-end training time by up to $2\times$ over baselines that use coarse models and achieves stable scalability on scenes that are otherwise infeasible for vanilla 3DGS. The main contributions of this work are summarized as follows:

- We identify load-balancing limitations in prior approaches and introduce a proxy that more closely correlates with fine-training runtime, enabling improved load balancing.

- We present LoBE-GS, featuring (i) load balance-aware scene partitioning for evenly distributed computational workloads, (ii) fast camera selection to minimize partition overhead, and (iii) visibility cropping and selective densification for accelerated fine-training.

- Extensive experiments show that LoBE-GS achieves a $2\times$ training speedup over existing methods while preserving rendering quality.

## 2 RELATED WORK

### 2.1 NOVEL VIEW SYNTHESIS

Given a set of captured images, novel view synthesis seeks to render photorealistic 3D scenes from previously unseen viewpoints. Neural Radiance Fields (NeRF) (Mildenhall et al., 2020) model radiance fields with an MLP and use volumetric ray marching to integrate color along camera rays. NeRF delivers high fidelity but incurs substantial training time and inference latency due to dense sampling and repeated neural evaluations. In contrast, 3D Gaussian Splatting (3DGS) (Kerbl et al., 2023) adopts Gaussian primitives, enabling differentiable rasterization for real-time rendering and training that often converges within minutes. While 3DGS yields strong quality, open issues include aliasing under wide baselines, semi-transparent geometry leakage, memory growth from millions of primitives, and robustness under sparse views or imperfect calibration. These advances and limitations motivate our design choices and evaluation, specially for large scale reconstruction.

## 2.2 LARGE-SCALE SCENE RECONSTRUCTION

For decades, reconstructing large-scale 3D scenes has been a central goal for researchers and engineers (Snavely et al., 2006; Agarwal et al., 2011). At city and regional scales, such reconstruction, especially for aerial views (Jiang et al., 2025; Tang et al., 2025), faces prohibitive memory demands and computational performance, motivating scalable training and rendering strategies.

**Distributed training approaches** train a unified model jointly across multiple GPUs. NeRF-XL (Li et al., 2024) shares NeRF parameters and activations across devices to maintain a single global model, executing multi-GPU volume rendering and loss computation with low inter-GPU communication, while DOGS (Chen & Lee, 2024) and Grendel-GS (Zhao et al., 2024) distribute Gaussian primitives via consensus or sparse all-to-all routing. CityGS-X (Gao et al., 2025) based on Grendel-GS proposes a scalable hybrid hierarchical representation with multitask batch rendering and training. It eliminate LoD merge-partition overhead while achieving efficient and geometrically accurate large-scale reconstruction. However, such systems typically require customized multi-GPU infrastructure to support frequent synchronization and communication, which limits their practicality on standard hardware setups.

**Divide-and-conquer approaches** partition a large scene into subregions, train submodels in parallel with multiple GPUs, and compose their outputs. Block-NeRF (Tancik et al., 2022) partitions a city into spatial blocks and assigns training views by camera position; Mega-NeRF (Turki et al., 2022a) decomposes space into grids and routes each pixel to the grids intersected by its ray; Switch-NeRF (Mi & Xu, 2023) learns the decomposition and routing end-to-end via a mixture-of-NeRF-experts. Within 3DGS representations, VastGS (Lin et al., 2024) introduces a progressive spatial partitioning strategy that divides a large scene into blocks and assigns training cameras and point clouds using an *airspace*-aware visibility criterion. Each block is optimized in parallel and subsequently fused to a seamless global 3DGS reconstruction. CityGS (Liu et al., 2025) leverages a coarse 3DGS prior to guide scene partitioning and parallel 3DGS submodel training, improving coherence and reconstruction quality across spatial partitions. They map unbounded scenes into a normalized unit cube and then partition the contracted scenes with a uniform grid for parallel training. However, most of the aforementioned works underemphasize load balancing of the submodels during partitioning, which limits parallel scalability. Moreover, CityGS loads the entire coarse model during the parallel stage, which is inefficient. To address these, LoBE-GS balances the 3DGS prior across submodels within each subregion and trains them efficiently in parallel.

## 2.3 EFFICIENT GAUSSIAN SPLATTING RECONSTRUCTION

As new 3DGS methods emerge, many research efforts target efficient 3D Gaussian Splatting reconstruction and rendering. With limited resources, 3DGS compression (Navaneet et al., 2024; Papantonakis et al., 2024) reduces on-disk storage, while Taming 3DGS (Mallick et al., 2024) addresses budget-constrained training and rendering via guided, purely constructive densification that steers growth toward high-contribution Gaussians. For large-scale scenes, level-of-detail (LoD) 3DGS representations enable efficient rendering (Ren et al., 2024; Kerbl et al., 2024). CityGaussianV2 (Liu et al., 2024) builds on CityGS (Liu et al., 2025) with an optimized parallel training pipeline that incorporates 2DGS for accurate geometric modeling. Momentum-GS (Fan et al., 2024) extends Scaffold-GS (Lu et al., 2024) to large-scale scenes by introducing scene momentum self-distillation and reconstruction-guided block weighting, allowing scalable parallel training with improved reconstruction quality. In this work, we focus on an efficient 3DGS reconstruction for large-scale scenes with coarse 3DGS prior and load-balanced parallel training.

## 3 ANALYSIS OF SCENE PARTITIONING AND LOAD BALANCING

In this section, we first show that existing scene partitioning strategies fail to achieve satisfactory load balancing during the fine-training stage. We then provide a principled analysis to identify a reliable proxy for estimating the per-block fine-training runtime.

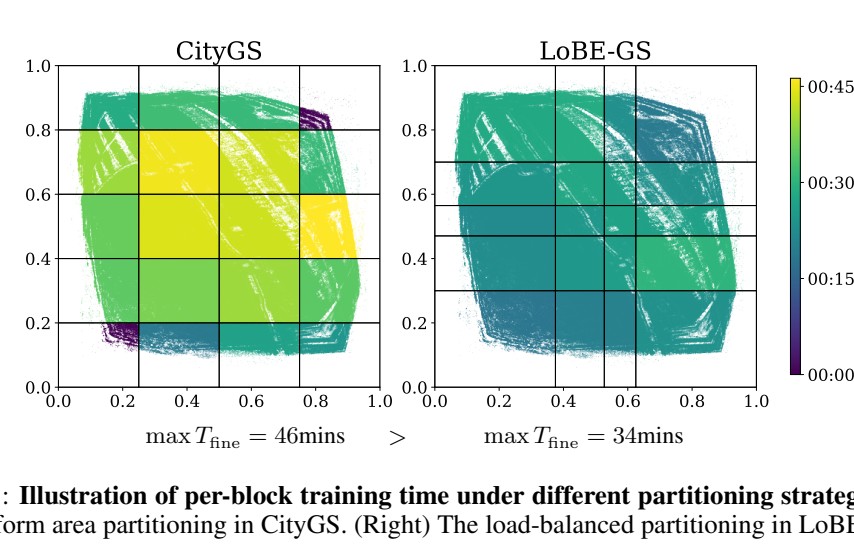

Figure 1: **Illustration of per-block training time under different partitioning strategies.** (Left) The uniform area partitioning in CityGS. (Right) The load-balanced partitioning in LoBE-GS.

### 3.1 IMPACT OF SCENE PARTITION ON LOAD BALANCING

Large-scale 3DGS pipelines typically adopt a *partition–and–merge* paradigm: the scene is divided into $B$ spatial blocks, each optimized independently in parallel, and then merged into a complete model. Some methods further employ a coarse-to-fine strategy, where a coarse model is trained first, followed by scene partitioning and parallel fine training before the final merging stage. Let $T_{\text{coarse}}$ denote the coarse-stage optimization time, $T_{\text{partition}}$ the partitioning time, and $T_{\text{fine}}^{(b)}$ the fine-stage runtime of block $b \in \{1, \ldots, B\}$. Assuming sufficient computational resources to run all fine-stage processes in parallel, the end-to-end runtime is defined as:

$$T_{\text{E2E}} = T_{\text{coarse}} + T_{\text{partition}} + \max_{b \in \{1, \ldots, B\}} T_{\text{fine}}^{(b)}. \tag{1}$$

Thus, an effective partitioning strategy must balance the workloads across blocks to minimize the runtime of the slowest block while maintaining reconstruction quality. Prior work has relied on heuristics such as equalizing *area*, *camera counts*, or *point counts*, yet their ability to predict fine-stage runtime were underexplored.

As a motivational example, consider the CityGS pipeline, which partitions the scene by equalizing block *areas* in contracted space. Figure 1 illustrates the fine-stage runtime per block on the *Building* dataset. Figure 1(a) shows that the strategy adopted by CityGS leads to significant load imbalance in fine-stage training. In contrast, Figure 1(b) shows that LoBE-GS achieves a more balanced runtime distribution by employing a different proxy. Similar patterns are observed across other datasets (see Appendix A.2), suggesting that existing heuristics are often suboptimal for actual fine-stage runtimes. As a result, they lead to skewed per-block runtimes and longer end-to-end runtime $T_{\text{E2E}}$.

### 3.2 RUNTIME CORRELATION WITH PER-BLOCK PREDICTORS

To address this, we analyze the correlation between candidate proxy variables and observed fine-stage runtimes to determine which predictors most accurately reflect the computational cost of each block. For each block $b$, we computed the Pearson correlation between its fine-stage runtime $T_{\text{fine}}^{(b)}$ (in minutes) and the following quantities, all available prior to fine-stage optimization:

- $A^{(b)}$: area of block $b$ in contracted space.
- $C^{(b)}$: number of cameras assigned to block $b$.
- $G_{\text{blk}}^{(b)}$: initial number of Gaussians inside block $b$ at the start of fine-stage optimization.
- $G_{\text{vis}}^{(b)}$: initial number of Gaussians visible across all cameras assigned to block $b$.
- $G_{\text{avg\_vis}}^{(b)} = G_{\text{vis}}^{(b)}/C^{(b)}$: initial average number of visible Gaussians per assigned camera.

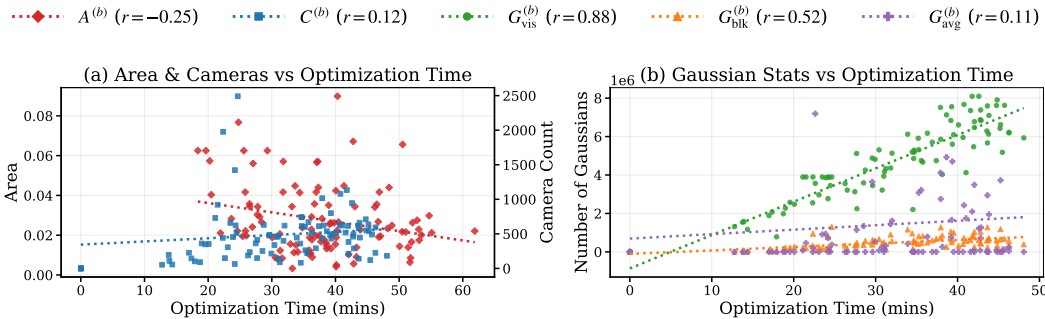

Figure 2: **Correlation between per-block training time and block-level statistics under CityGS's partitioning.** (a) Plots of block area $A^{(b)}$ and camera count $C^{(b)}$. (b) Plots of Gaussian-based measures ($G_{\text{blk}}^{(b)}$, $G_{\text{vis}}^{(b)}$, $G_{\text{avg\_vis}}^{(b)}$). $G_{\text{vis}}^{(b)}$ yields the strongest and most consistent correlation across datasets.

Figure 2 presents scatter plots across five representative datasets, *Building*, *Rubble*, *Residence*, *Sci-Art*, and *MatrixCity*, evaluated under fixed hardware and hyperparameters. Each point is color-coded by a candidate proxy variable, with fine-stage runtime on the x-axis and the corresponding proxy value on the y-axis. For each proxy, a dashed line of the same color hue is fit using linear regression. The legend also reports the Pearson correlation coefficients ($r$) between $T_{\text{fine}}^{(b)}$ and the respective block-level quantities.

The results indicate that the *area* proxy $A^{(b)}$, commonly adopted in prior works (Liu et al., 2025; 2024; Fan et al., 2024), exhibits relatively weak correlation with fine-stage runtime. Similarly, the per-block Gaussian count $G_{\text{blk}}^{(b)}$ shows minimal correlation, implying that considering only Gaussians physically contained within a block underestimates the effective optimization load. In contrast, the visibility-augmented measure $G_{\text{vis}}^{(b)}$ achieves the strongest and most consistent correlation across datasets, confirming its suitability as a reliable predictor of per-block training cost. Normalizing this quantity by camera count, resulting in $G_{\text{avg\_vis}}^{(b)}$, weakens the correlation, while the camera count alone, $C^{(b)}$, also used in previous studies (Chen & Lee, 2024; Yuan et al., 2025), exhibits only weak correlation. Overall, these findings suggest that balancing partitions by the number of initial visible Gaussians $G_{\text{vis}}^{(b)}$, as implemented in the proposed LoBE-GS, provides a more principled strategy than traditional equal-area or equal-camera approaches.

## 4 METHODOLOGY

Prior large-scale 3DGS systems have demonstrated strong results but continue to face challenges with load imbalance and training efficiency. To address these limitations, we propose LoBE-GS, a coarse-to-fine training framework where each block is fine-trained independently, following prior works (Liu et al., 2025; 2024). The overall pipeline is illustrated in Figure 3. Section 4.1 introduces *load balance-aware scene partition* that iteratively refines initial uniform cuts to minimize a proxy for fine-stage runtime. In Section 4.2, *fast camera selection* is proposed to improve efficiency over existing camera selection strategies. Finally, Section 4.3 describes *visibility cropping* and *selective densification*, two techniques that further reduce memory and computation costs during fine-training.

### 4.1 LOAD BALANCE-AWARE SCENE PARTITION

To mitigate load imbalance, we propose *load balance-aware scene partition* that minimizes maximum fine-training time $\max_b T_{\text{fine}}^{(b)}$ by leveraging proxy metrics $\max_b G_{\text{vis}}^{(b)}$, which exhibit strong correlation with fine-stage runtimes as analyzed in Section 3.2. For a grid partition with $B = m \times n$ blocks, given a coarse model $\mathcal{G}_{\text{coarse}}$ and a set of $c$ camera views, the objective is to optimize vertical

Figure 3: **Overview of our framework.** Our approach begins with training a coarse 3DGS model. Using our load balance–aware data partition, we optimize the grid cuts to achieve a more balanced division of the scene. We then apply visibility cropping and selective densification before and during the parallel fine-training stage, enabling faster and more efficient training. Finally, we prune regions outside each block and merge the results into a unified, high-quality model.

and horizontal cut positions $(\boldsymbol{v}, \boldsymbol{h})$ such that:

$$(\boldsymbol{v}^*, \boldsymbol{h}^*) = \arg \min_{\boldsymbol{v}, \boldsymbol{h}} \max_b G_{\text{vis}}^{(b)}(\boldsymbol{v}, \boldsymbol{h}), \tag{2}$$

where $\boldsymbol{v} = (v_1, \ldots, v_{m-1}) \in (0,1)^{m-1}$ and $\boldsymbol{h} = (h_1, \ldots, h_{n-1}) \in (0,1)^{n-1}$ denotes monotonically increasing cut positions in contracted space. The proxy $G_{\text{vis}}^{(b)}(\boldsymbol{v}, \boldsymbol{h})$ denotes the number of visible Gaussians in block $b = (i-1)(n+1) + j$ for $i \in \{1, \ldots, m\}$ and $j \in \{1, \ldots, n\}$, as defined by the corresponding cut boundaries $\mathcal{B}^{(b)}$ for the $i$-th row, $j$-th column block.

Since the computation of $G_{\text{vis}}^{(b)}(\boldsymbol{v}, \boldsymbol{h})$ is non-differentiable, we adopt Bayesian Optimization (BO) with a Gaussian Process (GP) surrogate for iterative cut refinement. We model the surrogate with a Matérn kernel and use the Expected Improvement (EI) acquisition function. The process begins with an initial uniform partition $(\boldsymbol{v}^{[0]}, \boldsymbol{h}^{[0]})$, where $(v_i^{[0]}, h_j^{[0]}) = (\frac{i}{m}, \frac{j}{n})$. To preserve ordering, each cut is constrained to move at most halfway toward its neighbors, i.e., $v_i \in [\frac{1}{2}(v_{i-1}^{[0]} + v_i^{[0]}), \frac{1}{2}(v_i^{[0]} + v_{i+1}^{[0]})]$ with $v_0 = 0$ and $v_m = 1$ (defined analogously for $h_j$). At each iteration $l$, BO proposes candidate cuts $(\boldsymbol{v}^{[l]}, \boldsymbol{h}^{[l]})$. The corresponding block regions are set to slightly enlarged grid cell, $\mathcal{B}^{(b)} = [v_{i-1}^{[l]} - \delta_v, v_i^{[l]} + \delta_v] \times [h_{j-1}^{[l]} - \delta_h, h_j^{[l]} + \delta_h]$, following prior works. Each block is then assigned a camera set $\mathcal{C}^{(b)}$ using standard view assignment strategies, and the number of visible Gaussians $G_{\text{vis}}^{(b)}$ is calculated. The GP surrogate is updated to fit the observed $\max_b G_{\text{vis}}^{(b)}(\boldsymbol{v}^{[l]}, \boldsymbol{h}^{[l]})$, and the best solution is tracked. After $L$ iterations, the best solution is returned. In practice, $L = 100$ and $(\delta_v, \delta_h) = (\frac{0.1}{m}, \frac{0.1}{n})$ yield satisfactory results, eliminating the need for reinitialization or nested search-space refinements.

## 4.2 FAST CAMERA SELECTION

*Camera selection* is performed to assign a subset of views $\mathcal{C}^{(b)}$ to each block for fine-training. The goal is to reduce per-block fine-training cost by discarding views with negligible coverage of the corresponding block region. This ensures that each block is optimized with only the most relevant views, improving efficiency without compromising reconstruction quality.

Despite its importance, prior studies often overlook the computational burden of this process, which can account for nearly half of the overall end-to-end runtime (see Section 5.3). For instance, given $M$ partitioned blocks and $N$ camera views, CityGS assigns cameras by computing the SSIM between the full coarse render and each per-block render, where the latter is obtained by filtering out Gaussians outside the block boundaries. This requires rendering every view for every block, resulting in at least $(M+1) \times N$ projections, which constitutes the main computational bottleneck.

To eliminate this overhead, we introduce *fast camera selection*, which reduces the computation to only $N$ projections. First, for each camera view, we compute the per-pixel depth $D$ using the $\alpha$-blending equation: $D = \sum_{i \in \mathcal{N}} d_i \alpha_i \prod_{j=1}^{i-1} (1 - \alpha_j)$, where $\mathcal{N}$ is the ordered set of points along the ray, $d_i$ the depth of point $i$, and $\alpha_i$ its opacity determined by covariance and opacity. The resulting

depth map is then back-projected into 3D space, forming a dense point cloud $\mathcal{P}^{(c)} = \{\boldsymbol{p}_{c,k} \mid k = 1, \ldots, K\}$, with $\boldsymbol{p}_{c,k} \in \mathbb{R}^3$ and $K$ denotes the total number of points for camera $c$. Next, for each camera $c$ and block $b$, we compute the visibility ratio of points inside the block:

$$V_{c,b} = \frac{1}{K} \sum_{k=1}^{K} \mathbb{1}[\boldsymbol{p}_{c,k} \in \mathcal{B}^{(b)}], \qquad (3)$$

where $\mathbb{1}$ denotes the indicator function, and $\mathcal{B}^{(b)}$ is the spatial region of block $b$. Finally, the assigned camera set for block $b$ is defined as $\mathcal{C}^{(b)} = \{c \mid V_{c,b} \geq \tau\}$, where $\tau$ is a predefined threshold (with $\tau = 0.15$) to prune views with negligible block coverage. This makes the procedure substantially faster, even enabling its use as a subroutine in BO, where the back-projection is computed once and reused throughout all iterations.

## 4.3 VISIBILITY CROPPING AND SELECTIVE DENSIFICATION

Prior coarse-to-fine 3DGS pipelines load the entire coarse model during per-block fine-training. This introduces both memory and runtime overhead. The memory overhead arises from storing the entire coarse model in GPU memory, while the runtime overhead arises from the Adam optimizer rather than rendering, as frustum culling already excludes non-visible points. Since Adam maintains momentum terms, it still updates parameters of all Gaussians, including those not observed by any camera in $\mathcal{C}^{(b)}$. Similar effects have also been observed in Mallick et al. (2024).

As fine-trained models are cropped before being merged into the final model, one naive solution is to retain only Gaussians strictly within each block $\mathcal{G}_{\text{blk}}^{(b)} = \{\boldsymbol{g} \in \mathcal{G}_{\text{coarse}} \mid \boldsymbol{g} \in \mathcal{B}^{(b)}\}$. However, this leads to degraded results due to over-pruning of Gaussians that lie outside block boundaries that remain visible in some views. To address this, we introduce *visibility cropping* that retain the visible Gaussians $\mathcal{G}_{\text{vis}}^{(b)} = \{\boldsymbol{g} \in \mathcal{G}_{\text{coarse}} \mid \boldsymbol{g} \text{ visible from some } c \in \mathcal{C}^{(b)}\}$ for each block prior to fine-training. This visibility-based filtering substantially reduces the number of Gaussians involved in optimization. In addition, since $G_{\text{vis}}^{(b)} = |\mathcal{G}_{\text{vis}}^{(b)}|$ must be recomputed at every BO iteration, we implement its evaluation entirely in NVIDIA Warp, achieving near-native CUDA performance and significantly reducing partition time. More implementation details are presented in Appendix A.1.

While *visibility cropping* preserves all visible Gaussians necessary for fine-training, it also includes those outside the block, i.e., $\mathcal{G}_{\text{vis}}^{(b)} \setminus \mathcal{G}_{\text{blk}}^{(b)}$, which are ultimately discarded prior to merging. Although retaining these Gaussians is essential to prevent quality degradation, they need not participate in densification. Motivated by this observation, we introduce *selective densification*, which restricts densification to Gaussians strictly within the block. This approach reduces the number of new Gaussians created during training, thereby lowering memory consumption and improving optimization efficiency, while maintaining per-block fidelity.

## 5 EXPERIMENTS

### 5.1 EXPERIMENTAL SETUP

**Datasets.** We conducted experiments on five large-scale scenes, including four real-world datasets and one synthetic dataset. For the real-world datasets, we used *Building* and *Rubble* from Mill19 (Turki et al., 2022b), and *Residence* and *Sci-Art* from UrbanScene3D (Lin et al., 2022). For the synthetic dataset, we adopted *Aerial*, which represents a small city region from MatrixCity (Li et al., 2023). Following prior work (Liu et al., 2025), all images in MatrixCity were resized to a width of 1600 pixels. For a fair comparison on real-world datasets, we downsampled all images by a factor of four, consistent with previous methods.

**Baselines.** We compare our framework against state-of-the-art large-scale 3DGS methods, including CityGS (Liu et al., 2025), VastGS (Lin et al., 2024), and DOGS (Chen & Lee, 2024). We also include 3DGS[†], which follows the original 3DGS pipeline but extends training to 60k iterations, sets the densification interval to 200 iterations, and applies densification until 30k iterations. For VastGS and DOGS, we directly adopt the metrics reported in DOGS paper, where VastGS was evaluated without appearance modeling. For runtime analysis, we use an unofficial implementation of VastGS

| Methods | MatrixCity-Aerial | | | | Residence | | | | Rubble | | | | Building | | | | Sci-Art | | | |
|---|---|---|---|---|---|---|---|---|---|---|---|---|---|---|---|---|---|---|---|---|
| | PSNR | SSIM | LPIPS | #GS | PSNR | SSIM | LPIPS | #GS | PSNR | SSIM | LPIPS | #GS | PSNR | SSIM | LPIPS | #GS | PSNR | SSIM | LPIPS | #GS |
| 3DGS | 23.67 | 0.735 | 0.384 | 9.7 | 21.44 | 0.791 | 0.236 | 6.6 | 25.47 | 0.777 | 0.277 | 6.1 | 20.46 | 0.720 | 0.305 | 6.4 | 21.05 | 0.837 | 0.242 | 3.7 |
| CityGS | 27.46 | 0.865 | 0.204 | 23.7 | **22.00** | **0.813** | 0.211 | 10.8 | 25.77 | **0.813** | 0.228 | 9.7 | 21.55 | 0.778 | 0.246 | 13.2 | **21.39** | 0.837 | 0.230 | 3.7 |
| Ours | **27.74** | **0.875** | **0.186** | 28.4 | 21.41 | 0.808 | **0.206** | 11.3 | **25.78** | 0.811 | 0.234 | 9.5 | **21.96** | **0.783** | 0.245 | 13.6 | 21.24 | **0.843** | **0.219** | 4.3 |

Table 1: **Quantitative comparison on MatrixCity, Mill19 and UrbanScene3D.** We report PSNR, SSIM, LPIPS, and the number of Gaussians (#GS, in millions).

| Methods | MatrixCity-Aerial | | | | Residence | | | | Rubble | | | | Building | | | | Sci-Art | | | |
|---|---|---|---|---|---|---|---|---|---|---|---|---|---|---|---|---|---|---|---|---|
| | PSNR | SSIM | LPIPS | #GS | PSNR | SSIM | LPIPS | #GS | PSNR | SSIM | LPIPS | #GS | PSNR | SSIM | LPIPS | #GS | PSNR | SSIM | LPIPS | #GS |
| VastGS | 28.33 | 0.835 | 0.220 | 10.3 | 21.01 | 0.699 | 0.261 | 3.7 | 25.20 | 0.742 | 0.264 | 4.7 | 21.80 | 0.728 | 0.225 | 5.6 | 22.64 | 0.761 | 0.261 | 3.5 |
| DOGS | 28.58 | 0.847 | 0.219 | 12.5 | 21.94 | 0.740 | 0.244 | 6.1 | 25.78 | 0.765 | 0.257 | 4.7 | 22.73 | 0.759 | **0.204** | 6.9 | 24.42 | 0.804 | 0.219 | 3.5 |
| Ours | **28.91** | **0.879** | **0.187** | 28.4 | **22.94** | **0.822** | **0.206** | 11.3 | **26.55** | **0.810** | 0.235 | 9.5 | **22.80** | **0.779** | 0.247 | 13.6 | **24.71** | **0.853** | 0.217 | 4.3 |

Table 2: **Color-corrected quantitative comparison on MatrixCity, Mill19 and UrbanScene3D.** We report color-corrected PSNR , SSIM , LPIPS , and the number of Gaussians (#GS, in millions).

(also without appearance modeling) to enable a fairer comparison of training efficiency. For consistency, we denote both variants as VastGS$^\dagger$ throughout our experiments. We do not report runtime results for DOGS, as its distributed training setup involves interconnect communication overhead, which is not directly comparable to our parallel but independent runtime setting.

**Metrics.** We evaluate reconstruction quality using PSNR, SSIM, and LPIPS. Since some prior works, such as DOGS and VastGS, apply color correction before computing these metrics, we also adopt the color-corrected versions to ensure fair comparison. In contrast, when comparing against 3DGS and CityGS, which do not apply color correction, we report the standard PSNR, SSIM, and LPIPS values.

**Efficiency metrics & runtime protocol.** We use $T_{\text{coarse}}$, $T_{\text{partition}}$, $\max T_{\text{fine}}$, and $T_{\text{E2E}}$ (as defined in Equation 1) as our efficiency metrics. For all runtime analysis presented in this paper, we adopt the same block configurations as CityGS: 36 blocks for *MatrixCity-Aerial*, 20 for *Building*, 20 for *Residence*, 9 for *Rubble*, and 9 for *Sci-Art*. All runtimes are measured on identical compute hardware, with detailed specifications provided in Appendix A.1.

### 5.2 QUANTITATIVE RESULTS

From Table 1 and Table 2, our method achieves competitive or superior reconstruction quality across datasets. Compared to CityGS, performance is largely on par, with modest gains ($\approx 1.0-1.02\times$) in PSNR/SSIM where applicable and consistently better LPIPS, at the cost of a slight PSNR drop on one dataset in exchange for improved perceptual quality. Compared to 3DGS$^\dagger$, we observe consistent improvements, typically $\approx 1.05-1.2\times$ higher PSNR/SSIM and up to $\sim 2\times$ lower LPIPS. With color-corrected metrics, our method also surpasses VastGS$^\dagger$ and DOGS on most datasets, leading in C-PSNR and C-SSIM. Overall, these results demonstrate parity with CityGS while clearly outperforming VastGS$^\dagger$, DOGS, and 3DGS$^\dagger$. Additional quantitative results are provided in Appendix A.7.

### 5.3 LOAD BALANCE AND RUNTIME ANALYSIS

As shown in Table 3, our method consistently achieves the lowest coarse-stage runtime and slowest-block fine-stage runtime across all datasets, yields the best partition time on two of three datasets, and achieves the best end-to-end runtime on MatrixCity-Aerial and UrbanScene3D; on Mill19, Notably, although our $T_{\text{E2E}}$ on Mill19 is slightly longer than the reported VastGS$^\dagger$ runtime (which omits $T_{\text{coarse}}$), our method delivers *higher reconstruction quality*—surpassing VastGS$^\dagger$ on PSNR, SSIM, and LPIPS (see Table 2)—highlighting a favorable quality–latency trade-off.

### 5.4 ABLATION STUDIES

To assess the contribution of each component in our framework, we conduct ablation experiments on three representative datasets: MatrixCity-Aerial, Residence, and Building. We evaluate different

| Methods | MatrixCity-Aerial | | | | Residence | | | | Rubble | | | | Building | | | | Sci-Art | | | |
|---|---|---|---|---|---|---|---|---|---|---|---|---|---|---|---|---|---|---|---|---|
| | $T_{coarse}$ | $T_{partition}$ | Max $T_{fine}$ | $T_{E2E}$ | $T_{coarse}$ | $T_{partition}$ | Max $T_{fine}$ | $T_{E2E}$ | $T_{coarse}$ | $T_{partition}$ | Max $T_{fine}$ | $T_{E2E}$ | $T_{coarse}$ | $T_{partition}$ | Max $T_{fine}$ | $T_{E2E}$ | $T_{coarse}$ | $T_{partition}$ | Max $T_{fine}$ | $T_{E2E}$ |
| 3DGS† | 01:50 | – | – | 01:50 | 01:22 | – | – | 01:22 | 01:10 | – | – | 01:10 | 01:30 | – | – | 01:30 | 00:40 | – | – | **00:40** |
| VastGS | – | 00:48 | 01:13 | 02:01 | – | 00:08 | 00:49 | **00:57** | – | 00:04 | 00:39 | 00:43 | – | 00:05 | 00:44 | 00:49 | – | 00:25 | 00:31 | 00:56 |
| CityGS | 00:52 | 01:39 | 01:00 | 03:31 | 00:43 | 00:31 | 01:22 | 02:36 | 01:06 | 00:09 | 01:14 | 02:29 | 00:59 | 00:21 | 01:06 | 02:26 | 00:42 | 00:08 | 00:45 | 01:35 |
| Ours | **00:38** | **00:16** | **00:30** | **01:24** | **00:26** | **00:08** | **00:30** | 01:04 | **00:23** | 00:05 | 00:41 | 01:09 | **00:25** | 00:08 | **00:30** | 01:03 | **00:16** | 00:05 | **00:26** | 00:47 |

Table 3: **End-to-end runtime comparison on MatrixCity, Mill19 and UrbanScene3D dataset.** For each dataset we report coarse time $T_{coarse}$, partition time $T_{partition}$, max fine time (Max $T_{fine}$), and total time $T_{E2E}$. A value of "–" indicates that the method does not include the corresponding stage.

| FCS | LB-SP | VC | SD | MatrixCity-Aerial | | | | Residence | | | | Building | | | |
|---|---|---|---|---|---|---|---|---|---|---|---|---|---|---|---|
| | | | | Max $T_{fine}$ | $T_{partition}$ | PSNR | SSIM | Max $T_{fine}$ | $T_{partition}$ | PSNR | SSIM | Max $T_{fine}$ | $T_{partition}$ | PSNR | SSIM |
| ✓ | | | | 01:25 | 00:14 | 27.77 | 0.874 | 01:01 | 00:04 | 21.71 | 0.810 | 01:06 | 00:03 | 21.73 | 0.784 |
| ✓ | ✓ | | | 01:24 | 00:16 | 27.80 | 0.875 | 00:54 | 00:07 | 21.49 | 0.810 | 00:54 | 00:08 | 21.84 | 0.781 |
| ✓ | | ✓ | | 00:52 | 00:14 | 27.75 | 0.874 | 00:47 | 00:04 | 21.66 | 0.810 | 00:45 | 00:03 | 22.15 | 0.787 |
| ✓ | ✓ | ✓ | | 00:47 | 00:16 | 27.77 | 0.875 | 00:36 | 00:07 | 21.58 | 0.811 | 00:34 | 00:08 | 21.51 | 0.762 |
| ✓ | | ✓ | ✓ | 00:32 | 00:14 | 27.76 | 0.874 | 00:33 | 00:04 | 21.84 | 0.813 | 00:39 | 00:03 | 22.11 | 0.787 |
| ✓ | ✓ | ✓ | ✓ | **00:30** | 00:16 | 27.74 | 0.875 | **00:30** | 00:07 | 21.41 | 0.808 | **00:30** | 00:08 | 21.96 | 0.783 |

Table 4: **Ablation on model components.** Evaluate the effectiveness of individual components: Fast Camera Selection (FCS), Load Balance-aware Scene Partition (LB-SP), Visibility Cropping (VC), and Selective Densification (SD).

combinations of four components: (1) **Fast Camera Selection (FCS)**, which accelerates camera-to-block assignment with negligible accuracy loss; (2) **Load Balance-aware Scene Partition (LB-SP)**, which redistributes Gaussians across blocks based on proxy load metrics to mitigate imbalance; (3) **Visibility Cropping (VC)**, which prunes invisible Gaussians to reduce optimization time; and (4) **Selective Densification (SD)**, which restricts densification to block regions. As shown in Table 4, LB-SP consistently reduces the worst-block fine-stage runtime $\max T_{fine}$: configurations with LB-SP always outperform otherwise identical ones without it. Moreover, enabling all four components halves the worst-block fine-stage runtime compared to the FCS-only baseline ($\sim$01:00 $\rightarrow$ $\sim$00:30), corresponding to a $\sim 2\times$ speedup in $\max T_{fine}$ and substantially improved end-to-end efficiency. These results highlight that LB-SP's workload rebalancing complements the per-block reductions of VC and SD, yielding the largest cumulative runtime gains when combined.

# 6 CONCLUSION

In this paper, we present LoBE-GS, which addresses load balancing and efficiency in the parallel training of 3DGS models. At the core of LoBE-GS is a computational-load proxy that enables an optimization for the scene partition of a coarse 3DGS model. We further introduce fast camera selection to accelerate the scene partitioning, as well as visibility cropping and selective densification to reduce loading in each block. LoBE-GS achieves up to $2\times$ training speedup over existing methods using coarse models for large-scale scene reconstruction while preserving the quality of the 3DGS models. In future work, we plan to experiment with larger and more complex scenes that would benefit from partitioning into a greater number of blocks for fine-training, and to explore the integration of level-of-detail (LoD) and 2DGS representations. We also plan to evaluate the framework on more diverse datasets, including those with sparse camera views in specific regions, and to investigate alternative partitioning strategies beyond the current grid-based approach.

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

# A  APPENDIX

## A.1  IMPLEMENTATION DETAILS

**Scene Partition.** Bayesian Optimization (BO) with Gaussian Process (GP) surrogate modeling (Section 4.1) is implemented using Ax (Olson et al., 2025) with BoTorch (Balandat et al., 2020) backend for GPU-accelerated optimization. Block load computation (Section 4.1), *fast camera selection* (Section 4.2), and *visibility cropping* (Section 4.3), are implemented in NVIDIA

Warp (Macklin, 2022), which enables kernel-based programming in Python with performance comparable to native CUDA. In preliminary benchmark on the *MatrixCity-Aerial* scene, the Warp implementation on a single GPU achieves speedups of approximately $450\times$ over sequential CPU code and $5\times$ over Numba-parallelized CPU code executed on 128 logical CPU cores. These performance improvements are enabled by low-level optimizations not exposed in PyTorch, including bitsets, atomics, and Warp tiles, with the latter providing functionality analogous to shared memory and cooperative groups in CUDA C++.

**3DGS Training.** The coarse-training stage employs the Sparse Adam optimizer to accelerate training, which has minimal impact on final performance. In contrast, the fine-training stage continues to use the standard Adam optimizer, as Sparse Adam was found to degrade performance in this setting. Aside from *selective densification*, fine-training details follows the standard vanilla 3DGS procedure (as in CityGS), with additional code-level optimizations through the *gsplat* library (Ye et al., 2025) and *fused-ssim* (Mallick et al., 2024) for SSIM loss evaluation.

**Experimental Setup.** For consistency, all CityGS runtimes reported in Section 3 are measured using a modified version of CityGS with *gsplat*, *fused-ssim*, and *visibility cropping* enabled. Moreover, since *selective densification* shortens per-block fine-training time, we disable it in LoBE-GS when reporting results in Section 3. A comparison against the unmodified CityGS with the full LoBE-GS pipeline (including *selective densification*) is provided in Figure A.1. In Section 5, since the official implementation of VastGS$^{\dagger}$ is unavailable, we report performance results based on an unofficial implementation available at `https://github.com/kangpeilun/VastGaussian`.

**System Configuration.** All experiments are conducted on a cluster consisting of 10 compute nodes, each equipped with 8 NVIDIA L40 GPUs and 128 logical CPU cores (Intel Xeon Platinum 8362), amounting to a total of 80 GPUs across the cluster. The fine-training stage is parallelized across blocks with one GPU per block, whereas all other stages are executed on a single GPU.

**Reproducibility.** Source code along with a pre-built Docker image will be released upon paper acceptance to ensure reproducibility. All reported runtimes are measured within the Docker environment to eliminate potential discrepancies caused by library mismatches or system-level variations.

**Declaration of LLM usage.** Large Language Models (LLMs) are only used for editing grammar.

## A.2 LOAD BALANCE ACROSS DATASETS

As shown in Figure A.1, our method yields a noticeably more uniform per-block workload distribution across the evaluated datasets. In particular, the load balance-aware partitioning combined with visibility cropping and selective densification systematically reduces the worst-case per-block fine-stage runtime, i.e., the slowest straggler blocks are much faster than under the baselines. This reduction in the tail of the runtime distribution leads to fewer stragglers and improved end-to-end efficiency. These gains are consistent across datasets, demonstrating the robustness of our partitioning strategy in mitigating workload skew.

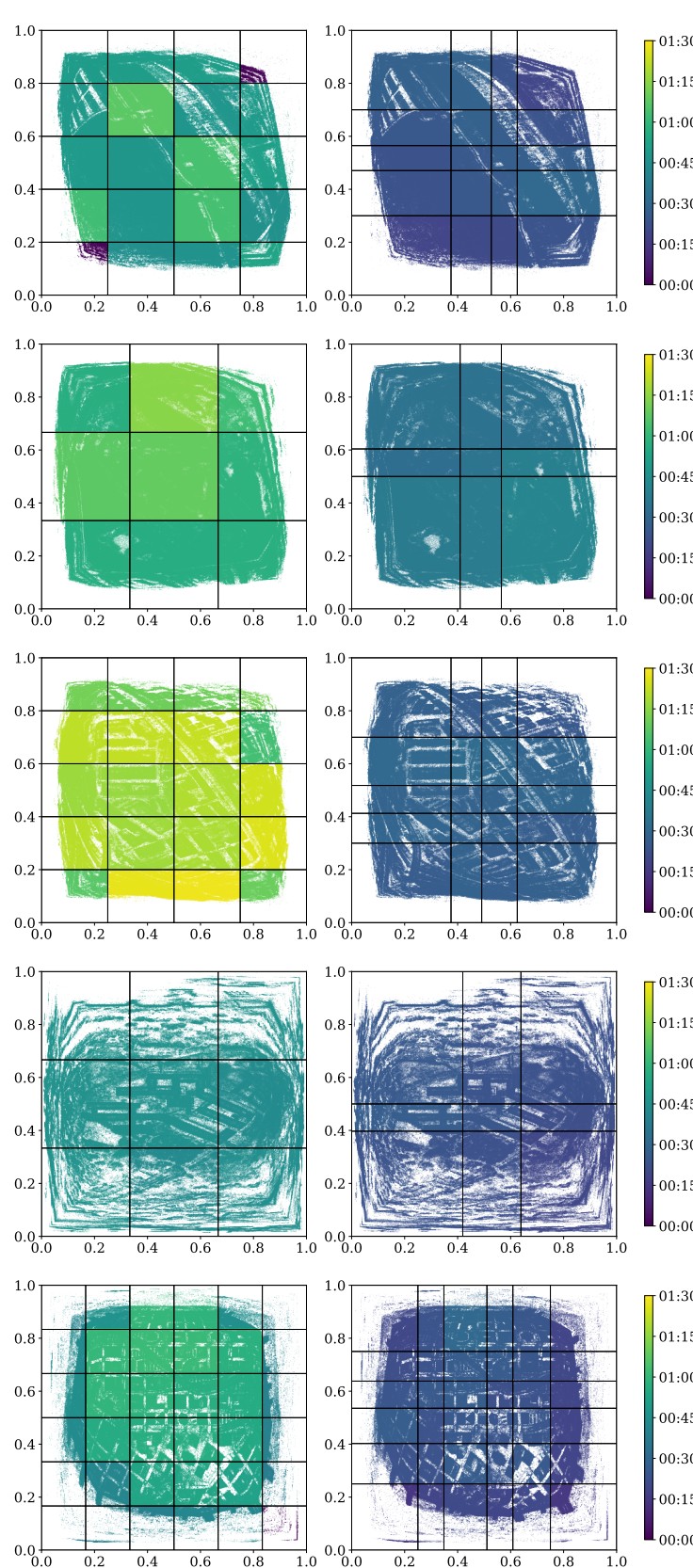

Figure A.1: Comparison of load balance and partitioning between CityGS (Left) and LoBE-GS (Right) across five datasets: *Building*, *Rubble*, *Residence*, *Sci-Art*, and *MatrixCity-Aerial*.

## A.3 COMPARISON AGAINST DISTRIBUTED 3DGS TRAINING SYSTEM

To assess how our framework compares to large-scale distributed training systems, we adopt GrendelGS as our primary baseline. GrendelGS is a distributed method that operates directly on explicit 3D Gaussians, the same representation used by LoBE-GS, which makes the comparison directly meaningful in terms of reconstruction quality, runtime, and deployment in existing explicit 3DGS viewers. Using the publicly available official implementation and following the settings described in the GrendelGS paper as closely as possible, we run GrendelGS on four NVIDIA L40 GPUs for 200k iterations with densification enabled up to 100k iterations. We report GrendelGS versus LoBE-GS on our benchmarks in terms of PSNR, SSIM, LPIPS, and end-to-end time, with quantitative results summarized in Table A.1. In the table below, LoBE-GS achieves substantially higher PSNR/SSIM and lower LPIPS while maintaining comparable end-to-end runtime on standard multi-GPU setups, without requiring tightly coupled distributed infrastructure (e.g., NVLink). We therefore position GrendelGS as a representative distributed explicit 3DGS baseline that emphasizes distributed training efficiency, whereas LoBE-GS targets a more favorable quality–versus–wall-clock trade-off under the same explicit representation.

| Method | Dataset | PSNR | SSIM | LPIPS | End-to-end time |
|--------|---------|------|------|-------|-----------------|
| GrendelGS (200k iters) | MatrixCity | 22.47 | 0.684 | 0.451 | 01:30 |
| LoBE-GS | MatrixCity | **27.74** | **0.875** | **0.186** | 01:24 |
| GrendelGS (200k iters) | Building | 20.14 | 0.680 | 0.326 | 00:59 |
| LoBE-GS | Building | **21.96** | **0.783** | **0.245** | 01:03 |
| GrendelGS (200k iters) | Rubble | 25.31 | 0.763 | 0.295 | 00:40 |
| LoBE-GS | Rubble | **25.78** | **0.811** | **0.234** | 01:09 |

Table A.1: Quality and runtime comparison between GrendelGS and LoBE-GS.

## A.4 EFFECT OF DIFFERENT PARTITION STRATEGIES

To isolate the effect of the partition strategy, we fix the coarse model and optimization setup across all methods: all variants are trained from the same coarse model and use the same visibility cropping and selective densification (VC+SD) configuration. We only vary the data partition strategy between VastGS, CityGS, and our load-balance–aware partitioning. As summarized in Table A.2, our method consistently achieves the lowest maximum fine-stage time (Max $T_{\text{fine}}$) on both Matrix-City and Building, while maintaining comparable or better reconstruction quality. On MatrixCity, attains slightly higher PSNR/SSIM and lower LPIPS than the alternative partition strategies, and on Building it matches or improves SSIM/LPIPS with only a marginal PSNR gap relative to CityGS. These results indicate that, under a shared coarse model and VC+SD setup, our partition strategy offers a more favorable quality versus Max-$T_{\text{fine}}$ trade-off.

| Dataset | Partition Strategy | #Blocks | PSNR | SSIM | LPIPS | $T_{\text{partition}}$ | Max $T_{\text{fine}}$ |
|---------|--------------------|---------|------|------|-------|-----------|-----------|
| MatrixCity | VastGS | 36 | 26.60 | 0.840 | 0.249 | 00:48 | 00:44 |
| MatrixCity | CityGS | 36 | 27.73 | 0.871 | 0.196 | 02:38 | 00:45 |
| MatrixCity | LoBE-GS | 36 | **27.74** | **0.875** | **0.186** | 00:16 | **00:30** |
| Building | VastGS | 20 | 21.65 | 0.765 | 0.265 | 00:05 | 00:35 |
| Building | CityGS | 20 | **22.01** | 0.782 | 0.249 | 00:23 | 00:40 |
| Building | LoBE-GS | 20 | 21.96 | **0.783** | **0.245** | 00:08 | **00:30** |

Table A.2: Comparison of different partition strategies in terms of reconstruction quality and runtime.

## A.5 SENSITIVITY TO COARSE 3DGS QUALITY

LoBE-GS assumes the availability of a coarse 3DGS model for partitioning and camera selection. To assess how sensitive the final performance is to the quality of this coarse prior, we vary the

number of coarse training iterations and re-run the full LoBE-GS pipeline on three datasets: MatrixCity, Building, and Rubble. In all cases, the partitioning and camera selection stages use the corresponding coarse model, while all other settings are kept fixed. Table A.3 reports the final reconstruction metrics after LoBE-GS for coarse training iterations of 10k, 20k, and 30k. Across all datasets, we observe that the resulting PSNR and SSIM vary only modestly ($\approx 0.3 - 0.6\,\mathrm{dB}$ PSNR and $\leq 0.02$ SSIM), while LPIPS consistently improves with more coarse iterations. At the same time, the number of Gaussians #GS grows moderately (e.g., from 25.5M to 28.4M on MatrixCity when increasing from 10k to 30k iterations). These trends suggest that LoBE-GS is relatively insensitive to the exact quality of the coarse model: even a shorter coarse stage (10k iterations) already provides a sufficiently accurate prior for partitioning and camera selection, and further refinement mainly yields incremental gains. Overall, this experiment confirms that LoBE-GS does not require a fully converged coarse 3DGS model. In practice, users can choose a shorter coarse training schedule to reduce preprocessing time, with only minor impact on the final reconstruction quality and a small change in the number of Gaussians. Moreover, because our fine-tuning stage is highly parallel and efficient, the final quality can be further improved by allocating more iterations to fine training rather than over-optimizing the coarse model. This flexibility allows practitioners to trade a shorter, cheaper coarse stage for slightly longer fine-tuning while maintaining strong reconstruction quality and overall training efficiency.

| Dataset | Coarse Iter. | Coarse | | | | Fine | | | |
|---|---|---|---|---|---|---|---|---|---|
| | | PSNR↑ | SSIM↑ | LPIPS↓ | #GS | PSNR↑ | SSIM↑ | LPIPS↓ | #GS |
| MatrixCity | 10k | 23.80 | 0.692 | 0.462 | 6.29 | 27.51 | 0.864 | 0.200 | 25.5 |
| | 20k | 24.55 | 0.730 | 0.414 | 8.00 | 27.34 | 0.867 | 0.197 | 25.7 |
| | 30k | 25.27 | 0.765 | 0.368 | 13.6 | 27.74 | 0.875 | 0.186 | 28.4 |
| Building | 10k | 18.98 | 0.635 | 0.394 | 3.24 | 21.49 | 0.762 | 0.259 | 10.0 |
| | 20k | 20.03 | 0.684 | 0.349 | 6.57 | 21.78 | 0.770 | 0.250 | 11.3 |
| | 30k | 20.41 | 0.711 | 0.324 | 9.66 | 21.96 | 0.783 | 0.245 | 13.6 |
| Rubble | 10k | 23.01 | 0.671 | 0.403 | 2.94 | 25.20 | 0.790 | 0.259 | 7.36 |
| | 20k | 24.21 | 0.726 | 0.342 | 5.61 | 25.35 | 0.791 | 0.254 | 8.27 |
| | 30k | 24.77 | 0.756 | 0.305 | 8.14 | 25.78 | 0.811 | 0.234 | 9.50 |

Table A.3: Sensitivity of LoBE-GS to the quality of the coarse 3DGS prior. We vary the number of coarse training iterations and report both coarse and final LoBE-GS reconstruction metrics on three datasets. #GS is the number of Gaussians in millions.

## A.6 ADDITIONAL CORRELATION ANALYSIS ACROSS DATASETS

In Section 3, we observed a strong correlation with $G_{\mathrm{vis}}^{(b)}$ when using the original CityGS pipeline combined with visibility cropping. In the fine-training stage, however, both visibility cropping and selective densification were enabled to further reduce the per-block load in LoBE-GS. To ensure that the correlation still remains strong under these settings, we additionally conducted experiments with both visibility cropping and selective densification enabled.

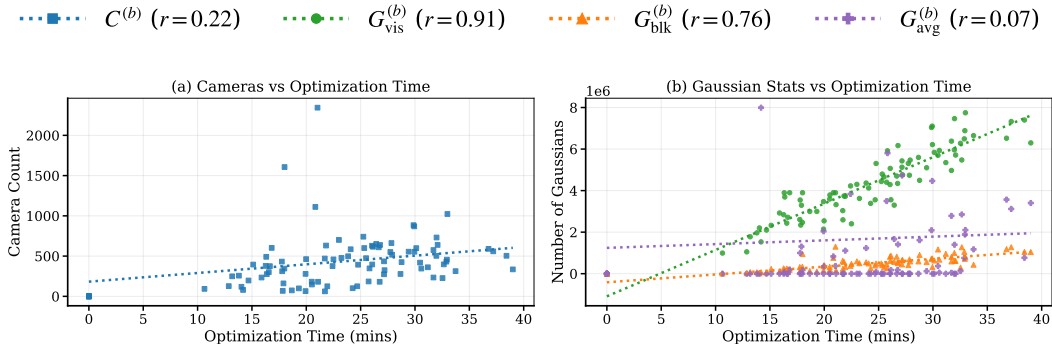

Figure A.2: **Correlation between per-block training time and block-level statistics under CityGS's partitioning with both visibility cropping and selective densification enabled.** (a) Plots camera count $C^{(b)}$. (b) Plots of Gaussian-based measures ($G_{\text{blk}}^{(b)}$, $G_{\text{vis}}^{(b)}$, $G_{\text{avg\_vis}}^{(b)}$). $G_{\text{vis}}^{(b)}$ yields the strongest and most consistent correlation across datasets even when selective densification is enabled.

## A.7 CPU/GPU PARTITION TIME ANALYSIS

Table A.4 compares load balance aware data partition times for CPU vs GPU implementations across five datasets.

| Methods | MatrixCity-Aerial | Residence | Rubble | Building | SciArt |
|---|---|---|---|---|---|
| LB-SP (CPU) | 00:47 | 00:18 | 00:03 | 00:15 | 00:10 |
| LB-SP (GPU) | 00:16 | 00:06 | 00:05 | 00:06 | 00:05 |

Table A.4: Partition time (`hh:mm`) comparison across CPU and GPU methods for five datasets.

