# OpenReview forum: "LoBE-GS: Load-Balanced and Efficient 3D Gaussian Splatting for Large-Scale Scene Reconstruction"
_ICLR.cc/2026/Conference — Submitted to ICLR 2026_

### Official Review · Reviewer_uL7E · 2025-10-29

**Soundness:** 2
**Presentation:** 2
**Contribution:** 3
**Rating:** 4
**Confidence:** 5

**Summary:**

LoBE-GS tackles challenges like load imbalance and inefficiencies in coarse-to-fine pipelines by introducing a depth-aware partitioning method, optimization to balance computational load, and lightweight techniques such as visibility cropping and selective densification. Evaluations show LoBE-GS achieves significantly faster training times compared to existing methods, without compromising reconstruction quality, and scales effectively to large, complex scenes.

**Strengths:**

1. The improvement of workload balance during parallel training is a considerable contribution towards a more efficient large-scale scene reconstruction.
2. Though the idea of visibility cropping has been implemented by V2 of CityGaussian, the exploration of the correlation between runtime and different factors is meaningful and inspiring.

**Weaknesses:**

1. The key contribution is the proposed partitioning strategy. To sufficiently validate the superiority, the authors should provide additional experiments on a specific baseline while only alternating the partitioning strategy, such as comparing with that of VastGaussian, CityGaussian, and Hierarchical 3DGS.  The authors should also illustrate the superiority of these baseline strategies.
2. The ablation only compares time cost, but the influence on the rendering quality is ignored, making the validation of module design incomplete.

**Questions:**

See weakness.

---

> ### Author Response · Authors · 2025-11-27
> **Reply to Reviewer uL7E**
>
> > ### W1. Partitioning-only comparisons
>
> Thank you for your structural comments. We conducted the experiments of the comparison to partition methods (VastGS / CityGS / LoBEGS) followed by our parallel fine-tuning method (w/ VC + SD). We have reported the results in the revised supplementary material (Section A.4), where we fix the coarse model and VC+SD configuration, and only vary the partitioning between VastGS, CityGS, and LoBEGS. The table below demonstrates the superiority of our strategy:
> - **LoBE-GS vs. CityGS**: We achieve comparable quality (27.74 vs 27.73  on MatrixCity) but with significantly lower partition time (16min vs 2hr 38min) and fine-tuning time (30min vs 45min).
> - **LoBE-GS vs. VastGS**: We outperform VastGS in both quality (+1.14 dB on MatrixCity) and total runtime.
> This isolates the benefit of our Load-Balanced Spatial Partitioning (LB-SP), proving it is more efficient than existing heuristics.
> Table A.2: Comparison of different partition strategies in terms of reconstruction quality and run-
> time.
>
> Table A.2: Comparison of different partition strategies in terms of reconstruction quality and runtime.
>
> | Dataset    | Partition Strategy | #Blocks | PSNR   | SSIM   | LPIPS  | Partition Time | Max Fine  Time |
> |:---------:|:------------------:|:-------:|:------:|:------:|:------:|:-----------:|:----------:|
> | MatrixCity| VastGS             | 36      | 26.60  | 0.840  | 0.249  | 00:48       | 00:44      |
> | MatrixCity| CityGS             | 36      | 27.73  | 0.871  | 0.196  | 02:38       | 00:45      |
> | MatrixCity| LoBE-GS            | 36      | **27.74** | **0.875** | **0.186** | 00:16       | 00:30      |
> | Building  | VastGS             | 20      | 21.65  | 0.765  | 0.265  | 00:05       | 00:35      |
> | Building  | CityGS             | 20      | **22.01** | 0.782  | 0.249  | 00:23       | 00:40      |
> | Building  | LoBE-GS            | 20      | 21.96  | **0.783** | **0.245** | 00:08       | 00:30      |
>
> > ### W2. Discussion on the rendering quality in the ablation study.
>
> Thank you. We added the rendering quality in the ablation table (Table 4). As shown in the table, all component combinations achieve very similar PSNR and SSIM scores across the MatrixCity-Aerial, Residence, and Building datasets (differences are within about 0.1-0.3 dB PSNR and 0.001-0.003 SSIM). Our efficiency components (LB-SP, VC, SD) drastically reduce runtime (from ~1.5h to ~0.5h) while maintaining consistent rendering quality across all datasets. This validates our design goal: to improve time efficiency and load balance without sacrificing the high-fidelity reconstruction characteristic of 3DGS.
>
>
> Table 4: Ablation on model components. Evaluate the effectiveness of individual components:
> Fast Camera Selection (FCS), Load Balance-aware Scene Partition (LB-SP), Visibility Cropping
> (VC), and Selective Densification (SD).
>
> | Dataset            | FCS | LB-SP | VC  | SD  | Max T_fine | T_partition | PSNR  | SSIM  |
> |--------------------|-----|-------|-----|-----|------------|-------------|-------|-------|
> | MatrixCity-Aerial  | ✓   |       |     |     | 01:25      | 00:14       | 27.77 | 0.874 |
> | MatrixCity-Aerial  | ✓   | ✓     |     |     | 01:24      | 00:16       | 27.80 | 0.875 |
> | MatrixCity-Aerial  | ✓   |       | ✓   |     | 00:52      | 00:14       | 27.75 | 0.874 |
> | MatrixCity-Aerial  | ✓   | ✓     | ✓   |     | 00:47      | 00:16       | 27.77 | 0.875 |
> | MatrixCity-Aerial  | ✓   |       | ✓   | ✓   | 00:32      | 00:14       | 27.76 | 0.874 |
> | MatrixCity-Aerial  | ✓   | ✓     | ✓   | ✓   | **00:30**  | 00:16       | 27.74 | 0.875 |
> | Residence          | ✓   |       |     |     | 01:01      | 00:04       | 21.71 | 0.810 |
> | Residence          | ✓   | ✓     |     |     | 00:54      | 00:07       | 21.49 | 0.810 |
> | Residence          | ✓   |       | ✓   |     | 00:47      | 00:04       | 21.66 | 0.810 |
> | Residence          | ✓   | ✓     | ✓   |     | 00:36      | 00:07       | 21.58 | 0.811 |
> | Residence          | ✓   |       | ✓   | ✓   | 00:33      | 00:04       | 21.84 | 0.813 |
> | Residence          | ✓   | ✓     | ✓   | ✓   | **00:30**  | 00:07       | 21.41 | 0.808 |
> | Building           | ✓   |       |     |     | 01:06      | 00:03       | 21.73 | 0.784 |
> | Building           | ✓   | ✓     |     |     | 00:54      | 00:08       | 21.84 | 0.781 |
> | Building           | ✓   |       | ✓   |     | 00:45      | 00:03       | 22.15 | 0.787 |
> | Building           | ✓   | ✓     | ✓   |     | 00:34      | 00:08       | 21.51 | 0.762 |
> | Building           | ✓   |       | ✓   | ✓   | 00:39      | 00:03       | 22.11 | 0.787 |
> | Building           | ✓   | ✓     | ✓   | ✓   | **00:30**  | 00:08       | 21.96 | 0.783 |

---

### Official Review · Reviewer_6K9G · 2025-10-29

**Soundness:** 3
**Presentation:** 2
**Contribution:** 3
**Rating:** 6
**Confidence:** 2

**Summary:**

This paper presents LoBE-GS, a load-balanced and efficient framework for large-scale 3D Gaussian Splatting (3DGS). While 3DGS has become a popular representation for real-time, high-fidelity 3D scene reconstruction, its scalability to large, unbounded scenes (e.g., city blocks) remains limited by memory and computational constraints. Existing divide-and-conquer strategies mitigate these issues by spatially partitioning scenes but introduce new bottlenecks due to load imbalance and inefficient coarse-to-fine training pipelines.

**Strengths:**

- The paper introduces a principled and quantitative approach to load balancing in large-scale 3DGS, leveraging the number of visible Gaussians as a proxy for computational demand—a novel insight that is both intuitive and empirically validated. The use of Bayesian Optimization for spatial partition refinement is also new in this context and provides a solid methodological contribution.
- LoBE-GS directly addresses a key bottleneck in scaling 3DGS to real-world environments. The proposed framework offers a tangible step toward city-scale Gaussian reconstruction, enabling more balanced and efficient large-scale modeling. The reduction of preprocessing time from hours to minutes is particularly impactful for practical deployments.

**Weaknesses:**

- Although LoBE-GS shows strong performance on multiple datasets, it would strengthen the paper to include comparisons against non-block-based large-scale approaches or hybrid systems (e.g., hierarchical or LoD 3DGS variants like CityGS-X or Momentum-GS). Moreover, an ablation isolating the contributions of each component (BO partitioning, camera selection, visibility cropping, selective densification) would provide clearer insight into their individual impacts.
- LoBE-GS assumes availability of a coarse 3DGS model for partitioning and camera selection. Discussion on how sensitive the method is to the quality of this coarse prior would be valuable, as suboptimal coarse models could degrade the partitioning accuracy.
- The method involves multiple implementation-specific optimizations (Warp kernels, GP-based BO). Publicly releasing code or detailing hyperparameter settings (e.g., BO iteration count, GP kernel choice) would improve reproducibility and adoption.

**Questions:**

See Weaknesses

---

> ### Author Response · Authors · 2025-11-27
> **Reply to Reviewer 6K9G - Part 1**
>
> > ### W1-1. Comparisons with non-block-based approaches
>
> Thank you for your comments. We compare LoBE-GS with GrendelGS, a state-of-the-art distributed training method for large-scale 3DGS reconstruction. As discussed in Related Work, our focus is on load-balanced partitioning and parallel training that scales across many GPUs without requiring specialized infrastructure. We ran GrendelGS on four NVIDIA L40 GPUs for 200k iterations with densification iteration to 100K and summarized the results below.
>
> Table A.1: Quality and runtime comparison between GrendelGS and LoBE-GS.
> | Method    | Dataset    | PSNR   | SSIM   | LPIPS  | End-to-End Time |
> |----------|------------|--------|--------|--------|------------------|
> | GrendelGS| MatrixCity | 22.47  | 0.684  | 0.451  | 01:30           |
> | LoBE-GS  | MatrixCity | **27.74** | **0.875** | **0.186** | 01:24           |
> | GrendelGS| Building   | 20.14  | 0.680  | 0.326  | 00:59           |
> | LoBE-GS  | Building   | **21.96** | **0.783** | **0.245** | 01:03           |
> | GrendelGS| Rubble     | 25.31  | 0.763  | 0.295  | 00:40           |
> | LoBE-GS  | Rubble     | **25.78** | **0.811** | **0.234** | 01:09           |
>
> Using the publicly available official implementation and following the settings described in the GrendelGS paper as closely as possible, we obtained the reconstruction metrics reported above on  the aforementioned hardware.  Our experiments show that LoBE-GS achieves substantially higher PSNR/SSIM and lower LPIPS while remaining easy to deploy on standard multi-GPU setups without tightly coupled distributed infrastructure (e.g., NVLink). We have discussed this experiment in more detail in the revised supplementary material (Section A.3).
>
> > ### W1-2. Component-wise ablation study
>
> Thank you for your comments.  We reported the ablation study to show the impact of each component in the revision (Table 4). As shown in the table below, we start from the FCS-only baseline, enabling LB-SP, VC, and SD progressively starting from a baseline (FCS only).
> -  Time Efficiency: The full configuration reduces the maximum fine-stage time by **~2–3×** (e.g., from 1h 25m to 30m on MatrixCity).
> - Quality: Critically, PSNR and SSIM remain essentially unchanged (fluctuations within ~0.3 dB and 0.003 SSIM).
> This confirms that each component contributes to a better efficiency-quality trade-off without degrading visual fidelity.
>
> Table 4: Ablation on model components. Evaluate the effectiveness of individual components:
> Fast Camera Selection (FCS), Load Balance-aware Scene Partition (LB-SP), Visibility Cropping
> (VC), and Selective Densification (SD).
>
> | Dataset            | FCS | LB-SP | VC  | SD  | Max T_fine | T_partition | PSNR  | SSIM  |
> |--------------------|-----|-------|-----|-----|------------|-------------|-------|-------|
> | MatrixCity-Aerial  | ✓   |       |     |     | 01:25      | 00:14       | 27.77 | 0.874 |
> | MatrixCity-Aerial  | ✓   | ✓     |     |     | 01:24      | 00:16       | 27.80 | 0.875 |
> | MatrixCity-Aerial  | ✓   |       | ✓   |     | 00:52      | 00:14       | 27.75 | 0.874 |
> | MatrixCity-Aerial  | ✓   | ✓     | ✓   |     | 00:47      | 00:16       | 27.77 | 0.875 |
> | MatrixCity-Aerial  | ✓   |       | ✓   | ✓   | 00:32      | 00:14       | 27.76 | 0.874 |
> | MatrixCity-Aerial  | ✓   | ✓     | ✓   | ✓   | **00:30**  | 00:16       | 27.74 | 0.875 |
> | Residence          | ✓   |       |     |     | 01:01      | 00:04       | 21.71 | 0.810 |
> | Residence          | ✓   | ✓     |     |     | 00:54      | 00:07       | 21.49 | 0.810 |
> | Residence          | ✓   |       | ✓   |     | 00:47      | 00:04       | 21.66 | 0.810 |
> | Residence          | ✓   | ✓     | ✓   |     | 00:36      | 00:07       | 21.58 | 0.811 |
> | Residence          | ✓   |       | ✓   | ✓   | 00:33      | 00:04       | 21.84 | 0.813 |
> | Residence          | ✓   | ✓     | ✓   | ✓   | **00:30**  | 00:07       | 21.41 | 0.808 |
> | Building           | ✓   |       |     |     | 01:06      | 00:03       | 21.73 | 0.784 |
> | Building           | ✓   | ✓     |     |     | 00:54      | 00:08       | 21.84 | 0.781 |
> | Building           | ✓   |       | ✓   |     | 00:45      | 00:03       | 22.15 | 0.787 |
> | Building           | ✓   | ✓     | ✓   |     | 00:34      | 00:08       | 21.51 | 0.762 |
> | Building           | ✓   |       | ✓   | ✓   | 00:39      | 00:03       | 22.11 | 0.787 |
> | Building           | ✓   | ✓     | ✓   | ✓   | **00:30**  | 00:08       | 21.96 | 0.783 |

---

> > ### Author Response · Authors · 2025-11-27
> > **Reply to Reviewer 6K9G - Part 2**
> >
> > > ### W2. Sensitivity to coarse-model quality
> >
> > Thank you for your comments. We conducted an experiment of the rendering quality against the coarse model train with 10k, 20k, and 30k iterations and then running the full LoBE-GS pipeline under a fixed fine-tuning budget (30k iterations). As shown in table, using a less-trained coarse model (fewer iterations) leads to a slight drop in the final reconstruction quality, but the effect is modest. This indicates that LoBE-GS is relatively insensitive to the exact quality of the coarse model: even a shorter coarse stage already provides a sufficiently accurate prior for partitioning and camera selection. Moreover, because our fine-tuning stage is highly parallel and efficient, the final quality can be further improved by allocating more iterations to fine training rather than over-optimizing the coarse model. This flexibility allows practitioners to trade a shorter, cheaper coarse stage for slightly longer fine-tuning while maintaining strong reconstruction quality and overall training efficiency. We have added this experiment in the revised supplementary material (Section A.5).
> >
> > Table A.3: Sensitivity of LoBE-GS to the quality of the coarse 3DGS prior. We vary the number of
> > coarse training iterations and report both coarse and final LoBE-GS reconstruction metrics on three
> > datasets. #GS is the number of Gaussians in millions.
> >
> > | Dataset    | Coarse Iter. | Coarse PSNR↑ | Coarse SSIM↑ | Coarse LPIPS↓ | Coarse #GS | Fine PSNR↑ | Fine SSIM↑ | Fine LPIPS↓ | Fine #GS |
> > |-----------|--------------|--------------|--------------|---------------|------------|------------|------------|-------------|----------|
> > | MatrixCity| 10k          | 23.80        | 0.692        | 0.462         | 6.29       | 27.51      | 0.864      | 0.200       | 25.5     |
> > | MatrixCity| 20k          | 24.55        | 0.730        | 0.414         | 8.00       | 27.34      | 0.867      | 0.197       | 25.7     |
> > | MatrixCity| 30k          | 25.27        | 0.765        | 0.368         | 13.6       | 27.74      | 0.875      | 0.186       | 28.4     |
> > | Building  | 10k          | 18.98        | 0.635        | 0.394         | 3.24       | 21.49      | 0.762      | 0.259       | 10.0     |
> > | Building  | 20k          | 20.03        | 0.684        | 0.349         | 6.57       | 21.78      | 0.770      | 0.250       | 11.3     |
> > | Building  | 30k          | 20.41        | 0.711        | 0.324         | 9.66       | 21.96      | 0.783      | 0.245       | 13.6     |
> > | Rubble    | 10k          | 23.01        | 0.671        | 0.403         | 2.94       | 25.20      | 0.790      | 0.259       | 7.36     |
> > | Rubble    | 20k          | 24.21        | 0.726        | 0.342         | 5.61       | 25.35      | 0.791      | 0.254       | 8.27     |
> > | Rubble    | 30k          | 24.77        | 0.756        | 0.305         | 8.14       | 25.78      | 0.811      | 0.234       | 9.50     |
> >
> >
> > > ### W3. Reproducibility and hyperparameter details
> >
> > Thank you. We will release the full codebase upon publication. In addition, we have reported the hyperparameters settings for our experiments in our method in Section 4.1 :
> > - BO iteration count: we use 100 iterations. (line 305)
> > - GP kernel: we adopt a Matérn kernel combined with the Expected Improvement acquisition function. (line 295-296)

---

### Official Review · Reviewer_YzRn · 2025-11-01

**Soundness:** 3
**Presentation:** 3
**Contribution:** 3
**Rating:** 4
**Confidence:** 5

**Summary:**

This paper proposes LoBE-GS, a load-balanced and efficient pipeline for large-scale 3D Gaussian Splatting (3DGS).

**Strengths:**

1. The paper identifies the “straggler block” issue in parallel 3DGS training and backs the visible-Gaussian proxy with correlation analyses and min-max balancing that directly reduces the worst-case fine-stage time.
2. Experiments span five scenes (four real, one synthetic), with both quality and runtime breakdowns; the “up to 2×” improvement is demonstrated alongside fair notes on color-alignment metrics and runtime components.

**Weaknesses:**

1. The core ideas (min-max balancing via a load proxy, linear-time camera assignment, pruning/controlled densification) are solid but incremental rather than conceptually radical for learning/reconstruction.
2. Can you record the specific number of Gaussians for each scene and analyze in detail the reasons behind the observed performance improvements?
3. I noticed that CityGS-X is discussed in the related work. This method eliminates the merge–partition overhead and performs parallel training across multiple GPUs, thereby naturally achieving load balance. Given that the main motivation of this paper is to improve load balancing through a better partitioning strategy, it would strengthen the paper if the authors could clarify how their approach differs from or improves upon CityGS-X.

In summary, my primary concern is about the novelty and technical advancement of the proposed approach.

**Questions:**

I have no more questions.

---

> ### Author Response · Authors · 2025-11-27
> **Reply to Reviewer YzRn - Part 1**
>
> > ### W1. The core ideas are solid but incremental rather than conceptually radical for learning/reconstruction.
>
> We thank the reviewer for recognizing the solidity of our core ideas. We revisit and address the issues in the large-scale scene reconstruction as a wall-clock–aware end-to-end problem.  Particularly, prior block-based methods (e.g., CityGS) largely overlook partition time, which may even exceed the training time. We propose the LoBE-GS with the key components:
> - a load proxy that directly targets max per-block runtime,
> - a linear-time camera assignment that makes this proxy usable at city scale, and
> - selective densification and pruning that Gaussians are added only where they reduce error per unit time.
>
> Together, they change the objectives and the reconstruction pipeline, yielding substantial reductions in end-to-end time with comparable quality across all benchmarks.
>
> > ### W2. Evaluation of the number of Gaussians for each scene and the reasons behind the performance improvements of LoBE-GS.
>
> Thank you for your comments. We have included the number of Gaussians for each method and scene in the revision (Table 1 and Table 2). As shown in the tables, LoBE-GS may produce a larger number of Gaussians than the previous methods due to its partitioning and parallel training strategy. We believe this explains the improvement in reconstruction quality. Despite the increase in Gaussian count, LoBE-GS still achieves faster training than the previous methods, demonstrating the efficiency of our method.
>
> Table 1: Quantitative comparison on MatrixCity, Mill19 and UrbanScene3D. We report PSNR,
> SSIM, LPIPS, and the number of Gaussians (#GS, in millions).
> | Method  | Dataset           | PSNR  | SSIM  | LPIPS | #GS (M) |
> |---------|-------------------|-------|-------|-------|---------|
> | 3DGS    | MatrixCity-Aerial | 23.67 | 0.735 | 0.384 | 9.7     |
> | 3DGS    | Residence         | 21.44 | 0.791 | 0.236 | 6.6     |
> | 3DGS    | Rubble            | 25.47 | 0.777 | 0.277 | 6.1     |
> | 3DGS    | Building          | 20.46 | 0.720 | 0.305 | 6.4     |
> | 3DGS    | Sci-Art           | 21.05 | 0.837 | 0.242 | 3.7     |
> | CityGS  | MatrixCity-Aerial | 27.46 | 0.865 | 0.204 | 23.7    |
> | CityGS  | Residence         | 22.00 | 0.813 | 0.211 | 10.8    |
> | CityGS  | Rubble            | 25.77 | 0.813 | 0.228 | 9.7     |
> | CityGS  | Building          | 21.55 | 0.778 | 0.246 | 13.2    |
> | CityGS  | Sci-Art           | 21.39 | 0.837 | 0.230 | 3.7     |
> | Ours    | MatrixCity-Aerial | 27.74 | 0.875 | 0.186 | 28.4    |
> | Ours    | Residence         | 21.41 | 0.808 | 0.206 | 11.3    |
> | Ours    | Rubble            | 25.78 | 0.811 | 0.234 | 9.5     |
> | Ours    | Building          | 21.96 | 0.783 | 0.245 | 13.6    |
> | Ours    | Sci-Art           | 21.24 | 0.843 | 0.219 | 4.3     |
>
> Table 2: Color-corrected quantitative comparison on MatrixCity, Mill19 and UrbanScene3D. We report color-corrected PSNR , SSIM , LPIPS , and the number of Gaussians (#GS, in millions).
> | Method | Dataset           | PSNR  | SSIM  | LPIPS | #GS (M) |
> |--------|-------------------|-------|-------|-------|---------|
> | VastGS | MatrixCity-Aerial | 28.33 | 0.835 | 0.220 | 10.3    |
> | VastGS | Residence         | 21.01 | 0.699 | 0.261 | 3.7     |
> | VastGS | Rubble            | 25.20 | 0.742 | 0.264 | 4.7     |
> | VastGS | Building          | 21.80 | 0.728 | 0.225 | 5.6     |
> | VastGS | Sci-Art           | 22.64 | 0.761 | 0.261 | 3.5     |
> | DOGS   | MatrixCity-Aerial | 28.58 | 0.847 | 0.219 | 12.5    |
> | DOGS   | Residence         | 21.94 | 0.740 | 0.244 | 6.1     |
> | DOGS   | Rubble            | 25.78 | 0.765 | 0.257 | 4.7     |
> | DOGS   | Building          | 22.73 | 0.759 | 0.204 | 6.9     |
> | DOGS   | Sci-Art           | 24.42 | 0.804 | 0.219 | 3.5     |
> | Ours   | MatrixCity-Aerial | 28.91 | 0.879 | 0.187 | 28.4    |
> | Ours   | Residence         | 22.94 | 0.822 | 0.206 | 11.3    |
> | Ours   | Rubble            | 26.55 | 0.810 | 0.235 | 9.5     |
> | Ours   | Building          | 22.80 | 0.779 | 0.247 | 13.6    |
> | Ours   | Sci-Art           | 24.71 | 0.853 | 0.217 | 4.3     |

---

> > ### Author Response · Authors · 2025-11-27
> > **Reply to Reviewer YzRn - Part 2**
> >
> > > ### W3. Discussion about CityGS-X
> >
> > Thank you for the comment. CityGS-X builds on GrendelGS, a distributed 3DGS training framework, and targets merge/partition overhead within an LoD-based representation. As discussed in Related Work, such distributed 3DGS approaches assume **tightly coupled, high-bandwidth multi-GPU infrastructure**: CityGS-X relies on **all-to-all Gaussian communication and DDP-based MLP synchronization**, which in practice requires specialized GPU–GPU interconnects (e.g., NVLink). On typical consumer or workstation setups, where GPUs lack direct high-speed links and communication falls back to PCIe, this substantially limits scalability and practical deployment. In contrast, LoBE-GS is designed to be easily deployable on standard multi-GPU systems or even on separate machines, as it **does not require any GPU-to-GPU communication**. Nevertheless, we agree that our description of CityGS-X could be improved. We have clarified the role and assumptions of CityGS-X in our revised Related Work (lines 118–120) and expanded the discussion of distributed 3DGS training in the supplementary material (Section A.3).

---

### Author Response · Authors · 2025-11-27
**General Response**

We thank all reviewers for their insightful comments and constructive feedback. We are encouraged that you found LoBE-GS to be a valuable and impactful contribution to large-scale 3DGS reconstruction. We appreciate the recognition of our novel methodological tools—specifically the spatial partition refinement and visibility-guided cropping—and our method’s ability to mitigate straggler-block issues, achieving up to 2× speed-up with strong empirical validation.
In this revision, we have addressed the concerns regarding baselines, ablations, and implementation details. Specifically, we have:
- Added a comparison with a non-block-based distributed method (GrendelGS) to demonstrate our advantages in standard multi-GPU setups.
- Expanded our ablation studies to include reconstruction quality metrics, confirming that our efficiency gains do not compromise visual fidelity.
- Analyzed the sensitivity of our method to the quality of the coarse 3DGS prior.
- Clarified implementation details and provided Gaussian counts for all scenes.

We have highlighted all revisions in the updated manuscript. Below, we address the specific concerns raised by each reviewer.

---

### Author Response · Authors · 2025-12-03
**Summary of Prior Discussions**

Dear Area Chair,

We note the recent reviewer-leak incident on OpenReview, and we sincerely appreciate your time and effort in contributing to the ICLR review process. To assist your evaluation, we provide below a summary of the previous discussions for your convenience.
In short, we have provided comprehensive and systematic responses to each of the reviewers’ comments. However, due to the shortened discussion period, we received no response, and no score changes were submitted.

Overall, the reviewers agree that LoBE-GS addresses an important and practical bottleneck in large-scale 3DGS by focusing on load-balanced, end-to-end runtime optimization (Reviewers YzRn, 6K9G, uL7E). They highlight as strengths: (i) the **visible-Gaussian load proxy and correlation analysis** as a principled way to model block runtime (Reviewers YzRn, 6K9G), (ii) the **load-balanced partitioning and BO-based refinement**, coupled with fast camera selection, as a solid methodological contribution for large-scale 3DGS (Reviewers 6K9G, uL7E), and (iii) **practical efficiency gains and scalability**, with reduced preprocessing time and up to ~2× faster training while maintaining high reconstruction quality on large, complex scenes (Reviewers 6K9G, uL7E).

***Reviewer YzRn*** raised concerns that the contributions may be incremental, asking for per-scene Gaussian counts and a clearer position to CityGS-X. In response, we clarified that LoBE-GS reframes large-scale 3DGS as a wall-clock–aware end-to-end problem, combining a visible-Gaussian load proxy with BO-based partition refinement, linear-time camera assignment, and selective densification, and we added detailed #GS statistics (Tables 1–2). We also explicitly contrasted LoBE-GS with CityGS-X, emphasizing that CityGS-X assumes tightly coupled multi-GPU infrastructure with all-to-all communication, whereas LoBE-GS targets standard multi-GPU or multi-machine setups without GPU–GPU communication.

***Reviewer 6K9G*** requested comparisons to non-block or hybrid large-scale methods, a component-wise ablation, a sensitivity study on the coarse prior, and clearer hyperparameter details. In response, we added a direct comparison to GrendelGS, a state-of-the-art distributed large-scale 3DGS method, showing that LoBE-GS attains clearly better PSNR/SSIM and lower LPIPS on MatrixCity, Building, and Rubble while retaining competitive end-to-end runtime.We also provided a detailed ablation over FCS, LB-SP, VC, and SD (≈2–3× reduction in max fine time with minor quality changes), a coarse-budget sensitivity study (10k/20k/30k iterations) showing modest impact on final quality, and explicit BO/hyperparameter settings (100 iterations, Matérn+EI), along with a commitment to release the code.

***Reviewer uL7E*** emphasized the importance of improved workload balance and found our runtime–factor analysis insightful, but asked both for a cleaner isolation of the partitioning strategy and for PSNR/SSIM to be reported in the ablation studies. In response, we performed partitioning-only comparisons (fixed coarse model + VC+SD, varying only VastGS / CityGS / LoBE-GS partition method), showing that LoBE-GS matches or slightly improves PSNR/SSIM while substantially reducing partition time and max fine time, especially on MatrixCity. We also extended the ablation table to include PSNR and SSIM alongside runtime, confirming that our efficiency gains (≈2–3× reduction in max fine time) come with only ~0.1–0.3 dB PSNR and ~0.001–0.003 SSIM variation.

Finally, as mentioned above, no reviewer followed up after our rebuttal, and did not adjust their score. Thus the current scores may under-reflect the clarifications and additional analyses we have provided. We hope this summary helps contextualize the discussion and the technical contributions of LoBE-GS when you form your recommendation.

Sincerely,
The Authors

---

### Meta-Review · Area_Chair_RFeq · 2026-01-06

**Summary:**

This submission proposes LoBE-GS, a framework aiming to improve the efficiency and load balance of 3D Gaussian Splatting (3DGS) for large-scale scenes. It identifies load imbalance in partitioned blocks and inefficiencies in coarse-to-fine pipelines as key bottlenecks. The proposed solutions include a depth-aware partitioning method to reduce preprocessing time, a Bayesian Optimization strategy to balance visible Gaussians (as a load proxy), and two techniques—visibility cropping and selective densification—to reduce per-block computational cost. Evaluations on several large-scale datasets report up to 2x faster end-to-end training compared to baselines like CityGS and VastGS, while claiming to maintain reconstruction quality. The work positions itself as a re-engineering of the large-scale 3DGS pipeline for better wall-clock performance on standard multi-GPU setups.

The decision to reject is based on the consensus regarding limited novelty and significant reproducibility concerns.

1.  **Limited Novelty:** The contribution is primarily an engineering system optimization (load balancing via a proxy) rather than a fundamental methodological advancement in 3D vision or representation learning. Balancing parallel workloads using a proxy is a standard systems engineering practice, making the novelty incremental for an ICLR submission.
2.  **Reproducibility & Implementation:** The core contribution—the non-uniform partitioning strategy—relies on a complex Bayesian Optimization loop. As noted by the AC and reviewers, the specific implementation details and code are crucial for verifying the validity of this approach. Without the code, the "black box" nature of the BO partitioning makes the results difficult to reproduce or verify.
3.  **Experimental Sufficiency:** While the authors demonstrate speedups, these are achieved through a combination of multiple engineering tricks (cropping, specific kernels) rather than a single distinct algorithmic breakthrough. It is difficult to disentangle the true source of the gains, and the comparisons against highly optimized distributed systems (like CityGS-X) remain a point of contention regarding the method's necessity.

Given the weighted feedback from high-confidence reviewers (both scoring 4) and the AC's assessment of the engineering/novelty gap, the paper does not meet the bar for acceptance.

**Reviewer Concerns:**

Reviewers raised significant concerns regarding the novelty and depth of the contribution:

1.  **Incremental Novelty (Reviewer YzRn):** The reviewer explicitly stated that the core ideas (min-max balancing via a load proxy, pruning) are "solid but incremental rather than conceptually radical." The method is viewed more as a collection of engineering optimizations rather than a fundamental advancement in learning or reconstruction.
2.  **Implementation & Reproducibility (Reviewer 6K9G):** The method relies heavily on implementation-specific optimizations (e.g., NVIDIA Warp kernels, specific GP-based BO settings). The reviewer raised concerns about reproducibility, noting that without code or extremely detailed hyperparameter settings for the BO partitioning, adoption would be difficult.
3.  **Baseline Comparisons (Reviewer uL7E & 6K9G):** Reviewers initially questioned the fairness of comparisons, asking for isolation of the partitioning strategy and comparisons to distributed systems like CityGS-X or GrendelGS. While the authors provided some additional data, the distinction between algorithmic gain vs. engineering gain remains blurred.
4.  **Complexity:** The reliance on a pre-trained coarse model adds a multi-stage complexity that may not justify the efficiency gains in all scenarios.

**Reviewer Scores:**

The reviewers' initial scores reflect a divided and cautious assessment of the submission's readiness:
*   **YzRn:** 4 (marginally below acceptance), Confidence 5
*   **6K9G:** 6 (marginally above acceptance), Confidence 2
*   **uL7E:** 4 (marginally below acceptance), Confidence 5

The scores indicate that two out of three reviewers, both with high confidence, placed the work below the acceptance threshold. The single positive score came with low confidence. Besides, no further scores are updated after rebuttal. The consensus leans towards the work being insufficient for acceptance in its current form.

---

### Decision · Program_Chairs · 2026-01-26

Reject